# Determinants of Quality of Life in Thai Caregivers of Dependent Older Adults: The Role of Health Promotion and Occupational Risks

**DOI:** 10.3390/ijerph22040578

**Published:** 2025-04-07

**Authors:** Sasithorn Thanapop, Bussarawadee Saengsuwan, Irniza Rasdi, Chamnong Thanapop

**Affiliations:** 1Department of Health Sciences, Sukhothai Thammathirat Open University, Nonthaburi 11120, Thailand; sasilau@gmail.com; 2Master of Public Health Program, School of Public Health, Walailak University, Nakhon Si Thammarat 80160, Thailand; bussarawadee.sae@gmail.com; 3Department of Environmental and Occupational Health, Faculty of Medicine and Health Science, Universiti Putra Malaysia, Serdang 43400, Malaysia; irniza@upm.edu.my; 4Research Centre of Data Science for Health Science, Walailak University, Nakhon Si Thammarat 80160, Thailand

**Keywords:** quality of life, caregivers, health promotion, occupational health risks, non-communicable diseases

## Abstract

Caregiving for dependent older adults presents significant health and occupational challenges, particularly in rural Thailand. This study examines predictors of health-related quality of life (HR-QOL) among in-home caregivers, focusing on health promotion, non-communicable diseases (NCDs) prevention, and occupational health practices. An analytic cross-sectional study was conducted among 701 caregivers across 23 districts in Nakhon Si Thammarat, southern Thailand. Data were collected through structured questionnaires assessing socio-demographics, health behaviors, working conditions, and HR-QOL using the Thai SF-36V2. Stepwise regression analysis identified key predictors of HR-QOL. Our findings indicate that most caregivers were female (81.5%), middle-aged (mean age: 47.7 years), and had moderate education levels. Over half experienced moderate exposure to occupational hazards, including physical, biological, ergonomic, and psychosocial risks, contributing to job strain, inadequate rest, and strained relationships. HR-QOL scores suggested that a significant proportion of caregivers had moderate to poor quality of life, particularly in physical health domains. Stepwise regression analysis showed that better self-reported health promotion behaviors and occupational hazard prevention practices were positively associated with HR-QOL, whereas the presence of NCDs had a negative impact. These findings highlight the need for targeted interventions to enhance caregivers’ well-being and inform public health strategies for strengthening community-based care systems.

## 1. Introduction

The global rise in life expectancy has resulted in an unprecedented increase in the proportion of older adults across all regions, with low- and middle-income countries experiencing the fastest growth in aging populations [1]. By 2050, two-thirds of the world’s population aged 60 years and older will reside in these regions. Recognizing the societal impact of aging, the United Nations declared the Decade of Healthy Aging (2021–2030), focusing on key priorities such as combating negative stereotypes about aging, creating age-friendly environments, improving healthcare services, and ensuring quality long-term care [2].

Thailand, as a rapidly aging society, exemplifies these challenges. Since 2005, the country has been categorized as an “aged society,” with older adults (≥60 years) comprising over 10% of the population [3]. This demographic shift necessitates robust community-based healthcare systems to address the diverse needs of older adults, particularly those who are dependent due to chronic illnesses, disabilities, or functional limitations [4]. To address this need, Thailand has implemented a long-term care (LTC) system that integrates formal care by healthcare professionals and informal care provided by family members or community volunteers. LTC for the elderly in Thailand is primarily categorized into two main service delivery models: (1) family and community-based services, where health agencies and local administrative organizations support community volunteers and families through budget allocation, social services, and knowledge dissemination; and (2) institutional care provided by public and private sectors, which focuses on standardized long-term care services [5].

In-home caregivers, community volunteers, or family caregivers play an indispensable role in supporting dependent older adults, often providing comprehensive assistance with daily activities, medical care, and emotional support [6]. However, caregiving can impose significant physical, psychological, and occupational burdens. Many European countries implement integrated LTC systems combining institutional and community-based services, supported by public funding and LTC insurance [7]. In contrast, ASEAN nations rely more on family and community-based care, with limited institutional facilities and emerging government support [8].

Thailand primarily relies on family and community-based elderly care, utilizing existing social capital. However, its caregiver support system is unstructured, especially regarding work-related conditions. Long working hours, insufficient rest, hazard exposure, and job strain negatively impact caregivers’ HR-QOL [9,10,11], despite a universal health coverage scheme that provides for general health services utilization. As a result, in-home caregivers lack systematic occupational health risk assessments and protections, increasing their vulnerability to health risks [12].

Despite the critical role of caregivers in sustaining LTC systems, research on their HR-QOL and the factors influencing it remains limited, particularly in rural and resource-constrained settings. Existing studies suggest that health promotion behaviors and occupational safety measures may mitigate these challenges, enhancing both caregiver well-being and the quality of care provided [13,14,15,16]. This research utilized the PRECEDE-PROCEED model’s initial phase [17], specifically the Predisposing, Reinforcing, and Enabling constructs to analyze determinants of health behaviors and environmental factors, especially workplace conditions to address this gap by identifying predictors of HR-QOL among in-home caregivers in rural Thailand, focusing on the interplay between health promotion, non-communicable diseases (NCDs) prevention, and occupational hazard prevention. The findings will provide valuable insights to inform public health strategies and enhance caregiver support systems in Thailand’s aging societies.

## 2. Materials and Methods

### 2.1. Study Design and Participants

This study employed an analytic cross-sectional design conducted in Nakhon Si Thammarat province, southern Thailand. The study population consisted of 9880 in-home caregivers providing care for dependent older adults. The sample size was calculated using the finite population proportion formula with a hypothesized proportion (p) of 0.3 and a 10% adjustment for potential non-responses, resulting in a required sample of 738 caregivers [18,19]. Using multi-stage random sampling, participants were selected proportionally from 23 districts and sub-districts. Eligible participants were in-home caregivers aged 18–59 years with at least six months of caregiving experience for dependent older adults. A total of 701 valid responses were obtained, yielding an effective response rate of 95.5%.

### 2.2. Data Collection and Measurements

Data were collected using a structured, four-part questionnaire designed to comprehensively assess various dimensions relevant to our study. The first section focused on sociodemographic characteristics, capturing information such as gender, age, marital status, educational attainment, the presence of NCDs, past occupations, monthly income, and caregiving-related subsidies. These characteristics were measured using a 10-item checklist [10,15].

The second section examined working conditions and hazard exposures through a 19-item checklist. This section included questions about working hours, caregiving experience, and exposure to physical, biological, ergonomic, and psychosocial hazards. Scores for hazard exposure were categorized into three levels: mild (0–4), moderate (5–9), and high (10–15) based on the class interval [9,13].

The third section evaluated self-health behaviors using a 17-item scale that covered two domains: health promotion and occupational hazard prevention behaviors. Responses were rated on a 5-point Likert scale, ranging from 1 (not at all) to 5 (extremely). The total score ranged from 17 to 85, with health behaviors categorized into good (≥80%), moderate (60–79.9%), or poor (<60%) [16,20,21].

The final section assessed HR-QOL using the Thai version of the SF-36 version 2.0 (SF-36V2), which is validated and reliable for use in the Thai population. This instrument evaluates eight dimensions of HR-QOL, with scores classified as poor, moderate, or good based on established cut-offs. The reliability of the physical and mental health summary scales was supported by Cronbach’s alpha values of 0.93 and 0.92, respectively [22,23,24].

The questionnaire’s validity was confirmed using the index of item-objective congruence (IOC = 0.96), and internal consistency was assessed with Cronbach’s alpha (0.80 for self-health behaviors). Data collection was conducted through face-to-face sessions across all sub-districts, with researchers providing explanations of the study’s objectives and procedures to participants. Each session took approximately 30 min to complete, ensuring that participants fully understood the questions and responded accurately.

### 2.3. Statistical Analysis

Data were analyzed using SPSS version 23.0 (IBM, Singapore). Descriptive statistics (means, standard deviations (SD), and frequencies) were used to summarize participant characteristics and scores. Bivariate correlations between independent variables (socio-demographics, working conditions, hazard exposures, and self-health behaviors) and HR-QOL were assessed using a point-biserial correlation and Pearson’s chi-squared tests. Variables with a *p*-value < 0.25 were included in the multivariate analysis [25]. Stepwise regression was employed to identify significant predictors of HR-QOL. The final model controlled for gender and age, with statistical significance set at *p* < 0.05.

## 3. Results

### 3.1. The Socio-Demographics, Working Conditions, and Health of In-Home Caregivers

A total of 701 in-home caregivers participated in this study, with the majority being female (81.5%). The mean age of participants was 47.7 years (SD = 7.9), and 71.5% were aged 45 years or older. Most participants were married (71.2%) and had completed primary or high school education (91.2%). About 31.4% of caregivers reported having NCDs. The majority (71.8%) earned less than 15,000 Thai Baht per month, and only 30.1% received caregiving-related subsidies.

Regarding their caregiving workloads, most caregivers provided care for one or two older adults (69.5%) and worked an average of 11.9 h per day (SD = 10.0), with 53.8% working more than 8 h daily. The average caregiving experience was 4 years (SD = 2.8), with 53.9% reporting 1–3 years of experience (Table 1).

### 3.2. Hazard Exposures Among In-Home Caregivers

In-home caregivers faced multiple occupational hazards across physical, biological, ergonomic, and psychosocial domains (Table 2). Physical hazards included inadequate lighting (19.3%), poor air circulation (20.5%), and disturbing noise (17.8%). Biological hazards were prevalent, with 77.3% reporting contact with excreta and wound drainage, and 69.0% encountering skin infections. Ergonomic risks were significant, as 80.0% assisted with locomotion and 61.1% experienced prolonged standing or sitting, increasing the risk of musculoskeletal disorders.

Psychosocial stressors were also prominent. Over half of the caregivers reported job strain (58.6%), high service quality expectations from relatives (58.3%), and inadequate rest or sleep (55.8%). Although less frequent, strained relationships with older relatives (12.3%) and the complexities of caregiving roles (10.8%) were noted. These exposures highlight the challenging working conditions faced by caregivers, which can adversely affect their health and well-being.

### 3.3. The Health-Related Quality of Life (HR-QOL) of In-Home Caregivers

The HR-QOL scores revealed significant challenges, particularly in physical health domains (Table 3). Most caregivers reported moderate HR-QOL (58.9%), while 23.0% had poor HR-QOL. Physical functioning was a significant concern, with nearly half (46.6%) reporting poor or moderate levels. Role-physical limitations were prevalent, with 27.0% and 35.0% reporting poor and moderate scores, respectively. General health was skewed towards poor (39.7%) and moderate (28.4%) levels.

In contrast, psychological health domains were relatively better. Social functioning was rated as good by 42.9% of caregivers, while vitality scores were evenly distributed across poor (33.0%), moderate (32.8%), and good (34.2%) categories. Mental health scores showed a mixed distribution, with 29.4% reporting poor mental health.

### 3.4. Key Predictors of HR-QOL

Stepwise regression analysis identified several significant predictors of HR-QOL (Table 4). Positive predictors included health promotion behaviors (β = 0.269, *p* < 0.001) and occupational hazard prevention practices (β = 0.118, *p* < 0.01). Conversely, the presence of NCDs was a significant negative predictor (β = −0.132, *p* < 0.001). The final regression model explained 38.4% of the variance in HR-QOL (R² = 0.384).

## 4. Discussion

### 4.1. Socio-Demographic and Occupational Challenges

This study highlights the demographic profile of in-home caregivers in southern Thailand, where the majority are middle-aged females with limited education and low income. These findings align with previous research indicating that caregiving remains a female-dominated field, often involving individuals from lower socioeconomic backgrounds [10,14,26,27,28,29].

The monthly income of caregivers is typically low, with most earning less than 15,000 Thai baht. Additionally, caregivers work long hours, averaging 11.9 h per day, and often lack formal employment benefits, such as work-related subsidies. In-home caregivers, in particular, frequently face excessive workloads; more than half report working over 8 h daily—exceeding the standard working hours of healthcare professionals in institutional settings. These conditions contribute to a range of adverse outcomes, including physical exhaustion, psychological distress, strained familial relationships, and an increased risk of compromised quality of care for care recipients. Limited income and restricted access to employment-related benefits (reported by only 30.1% of caregivers) indicate significant financial strain, which may adversely affect their well-being and job satisfaction [30]. These challenging circumstances are likely to exacerbate stress levels and reduce opportunities for self-care, ultimately negatively impacting their HR-QOL. Thus, policy reforms and employer-driven initiatives are essential to ensure fair compensation and the provision of social safety nets for caregivers.

In addition, a significant proportion (31.4%) of caregivers reported having NCDs, which may reduce their physical capacity and increase their healthcare needs [31]. Long working hours and irregular schedules among caregivers have been consistently highlighted in previous research [32]. These conditions contribute significantly to caregiver fatigue and burnout [13]. Implementing alternative scheduling models or providing overtime compensation could help mitigate these challenges. Furthermore, community-based LTC systems have been shown to enhance caregiver experiences by offering social support from local authorities and primary care services [14].

### 4.2. Hazard Exposures in Caregiving Roles

Caregivers face significant occupational hazards, including physical, biological, ergonomic, and psychosocial risks. The prevalence of biological exposures, such as contact with excreta and wound drainage, reflects inadequate infection control practices, consistent with findings from studies in resource-limited settings [33,34,35]. Ergonomic risks, such as prolonged standing and assisting with locomotion, highlight the physical demands of caregiving, which increase susceptibility to musculoskeletal disorders [34,36]. Psychosocial stressors, including job strain and inadequate rest, further exacerbate caregiver burden. These hazards are known to compromise caregivers’ physical and mental well-being, potentially leading to burnout and a reduced quality of care provided to older adults [10,34].

### 4.3. Health-Related Quality of Life and Key Predictors

The study found that HR-QOL among caregivers was predominantly moderate, with significant challenges in physical health domains. This is consistent with previous studies indicating that caregiving often takes a toll on physical functioning due to demanding workloads and insufficient recovery time [34,35]. Female gender was a positive predictor of higher quality of life, which may be attributed to socio-cultural norms in Thailand that traditionally assign caregiving and domestic responsibilities to women, who often hold more favorable attitudes toward older adult care [8]. This finding aligns with the national LTC strategy, which emphasizes the mobilization of social capital and community-based support—particularly through trained village health volunteers—within resource-limited settings [5].

The identification of health promotion behaviors as a positive predictor of HR-QOL underscores the importance of self-care practices in mitigating caregiving-related stress [37,38]. Similarly, occupational hazard prevention was associated with better HR-QOL, highlighting the need for targeted interventions to improve workplace safety and reduce exposure to hazards [39]. Conversely, the presence of NCDs emerged as a significant negative predictor, emphasizing the dual burden of caregiving and managing chronic health conditions [34,35].

### 4.4. Implications for Policy and Practice

Our findings suggest the need for comprehensive support programs that address both the physical and psychological demands of caregiving. Occupational health management should prioritize the development of risk mitigation skills targeting biological and ergonomic hazards, which are the primary contributors to occupational risks affecting health-related quality of life (HR-QOL). Local health authorities should integrate occupational health and safety measures into primary healthcare services for in-home elderly caregivers, akin to existing initiatives for informal workers. This includes implementing workplace risk assessment programs, offering training on hazard identification and risk evaluation, promoting safe caregiving practices in home settings, and encouraging family collaboration in areas such as work scheduling, rest planning, and the effective use of local resources.

Policies should prioritize fair compensation, equitable access to healthcare, and comprehensive training in infection control and ergonomic practices [13,14]. Community-based interventions—such as caregiver support groups and stress management programs—can further enhance caregiver well-being, particularly in contexts where social capital is embedded in community and cultural structures, as in Thailand [37,38]. Moreover, integrating health promotion initiatives into daily caregiving routines may encourage sustainable self-care behaviors, thereby improving caregivers’ HRQOL and enhancing the overall quality of care provided to older adults [16,38].

### 4.5. Limitations

This study has several limitations. First, its cross-sectional design limits the ability to establish causal relationships between predictors and HR-QOL. Longitudinal studies are needed to better understand the dynamic interplay between caregiving roles, health behaviors, and HR-QOL over time. Second, the reliance on self-reported data introduces the potential for recall bias and social desirability bias, which may have influenced participants’ responses. Lastly, the study was conducted in a single province, which may limit the generalizability of findings to other regions with differing caregiving contexts and resources. Future research should explore these dynamics in diverse settings and incorporate objective measures of caregiver health to strengthen the evidence base.

## 5. Conclusions

This study highlights the significant challenges faced by in-home caregivers in southern Thailand, including long working hours, exposure to occupational hazards, and the burden of managing NCDs. These factors negatively impact their HR-QOL, particularly in physical health domains. However, health promotion behaviors and occupational hazard prevention practices emerged as critical predictors of better HR-QOL. Targeted interventions, including caregiver training, workplace safety improvements, and health promotion programs, are essential to enhance caregiver well-being and sustain community-based long-term care systems.

## Figures and Tables

**Table 1 ijerph-22-00578-t001:** Socio-demographics, working conditions and hazards exposures level, and self-health behaviors of the in-home caregivers (*n* = 701).

Characteristics	*n*	%
Socio-demographic		
Gender		
Male	130	18.5
Female	571	81.5
Age (year) (Mean 47.7, SD 7.9, Max 59, Min 20)		
<45	200	28.5
≥45	501	71.5
Marital status		
Married	499	71.2
Single/widow/separate	202	28.8
Education		
Primary to high school	639	91.2
Bachelor’s degree	62	8.8
Non-communicable diseases		
Presence	220	31.4
Absence	481	68.6
Past/main occupations		
Agriculture	389	55.5
Employee/Civil servant	312	44.5
Income (Thai-baht per month *)		
≤15,000	503	71.8
>15,000	198	28.2
Caregiving-related subsidies		
Subsidy	211	30.1
No subsidy	490	69.9
Working conditions		
Number of caregiving older		
1–2	487	69.5
≥3	214	30.5
Working hour per day (Mean 11.9, SD 10.0, Max 24, Min 1)		
1–7	324	46.2
≥8	377	53.8
Working day per month (Mean 20.9, SD 10.6, Max 30, Min 4)		
1–14	235	33.5
≥15	466	66.5
In-home care experience (year) (Mean 4.0, SD 2.8, Max 15, Min 1)		
1–3	378	53.9
4–6	221	31.5
>6	102	14.6
Hazards exposure level (Mean 6.8, SD 3.0, Max 12, Min 0)		
Low	183	26.1
Moderate	373	53.2
High	145	20.7
Self-health behaviors		
Health promotion (Mean 33.3, SD 5.6, Max 50, Min 20)		
Poor	180	25.7
Moderate	423	60.3
Good	98	14.0
Occupational hazard prevention (Mean 26.9, SD 5.3, Max 35, Min 7)		
Poor	76	10.8
Moderate	300	42.8
Good	325	46.4

* 1 USD = 34.26 Thai-baht (Accessed 9 November 2024).

**Table 2 ijerph-22-00578-t002:** Prevalence of Occupational Hazard Exposures Among In-Home Caregivers (*n* = 701).

Hazards Exposure	*n*	%
Physical exposure		
(1) Inadequate lighting conditions	135	19.3
(2) Limited air ventilation	144	20.5
(3) Disturbance noise	125	17.8
(4) Multi-story construction	92	13.1
Biological exposure		
(5) Direct contact with excreta and wound drainage	542	77.3
(6) Direct contact with aspirate	347	49.5
(7) Direct contact with the skin infection	484	69.0
Ergonomics exposure		
(8) Assisting with side-lying positioning	546	77.9
(9) Assistance with maintaining postural control and locomotion	561	80.0
(10) Engaging in prolonged sitting or standing	428	61.1
Psychosocial exposure		
(11) Job strain	411	58.6
(12) Service quality expectations from ageing relatives	409	58.3
(12) Inadequate rest or sleep	391	55.8
(14) Strained relationships with older relatives	86	12.3
(15) Coping with the demands of a complex caregiving role	76	10.8

**Table 3 ijerph-22-00578-t003:** Health-Related Quality of Life (HR-QOL) Levels Among In-Home Caregivers by Domain (*n* = 701).

Domains	HR-QOL Levels: n (%)		
	Poor	Moderate	Good
** *Overall* **	161 (23.0)	413 (58.9)	127 (18.1)
** *Physical health* **			
Physical functioning	327 (46.6)	183 (26.1)	191 (27.2)
(Mean 64.5, SD 20.2, Max 100, Min 0)			
Role limitations due to physical health	189 (27.0)	245 (35.0)	267 (38.1)
(Mean 71.5, SD 20.5, Max 100, Min 0)			
Bodily pain	196 (28.0)	273 (38.9)	232 (33.1)
(Mean 73.0, SD 20.0, Max 100, Min 0)			
General	278 (39.7)	199 (28.4)	224 (32.0)
(Mean 66.3, SD 19.1, Max 100, Min 0)			
** *Psychological health* **			
Social functioning	73 (10.4)	327 (46.6)	301 (42.9)
(Mean 77.5, SD 16.7, Max 100, Min 0)			
Vitality	231 (33.0)	230 (32.8)	240 (34.2)
(Mean 68.2, SD 18.3, Max 100, Min 12.5)			
Role emotion	231 (30.4)	141 (20.1)	347 (49.5)
(Mean 75.1, SD 23.6, Max 100, Min 0)			
Mental health	206 (29.4)	200 (28.5)	295 (42.1)
(Mean 72.1, SD 17.2, Max 100, Min 15)			

**Table 4 ijerph-22-00578-t004:** Stepwise Regression Analysis of Predictors of Health-Related Quality of Life (*n* = 701).

Model		Health-Related Quality of Life				
		R^2^	B	SE	Beta	t
1	(Constant)		1619.401	95.214		17.008 **
	Health promotion	0.329	25.957	2.816	0.329	9.217 **
2	(Constant)		1685.004	95.773		17.594 **
	Health promotion		25.220	2.795	0.320	9.024 **
	NCDs presence	0.357	−130.632	33.652	−0.138	−3.882 **
3	(Constant)		1520.293	106.329		14.298 **
	Health promotion		21.248	3.002	0.269	7.077 **
	NCDs presence		−124.234	33.443	−0.131	−3.715 **
	Occupational hazard prevention	0.377	10.967	3.175	0.132	3.454 *
4	(Constant)		1481.719	107.544		13.778 **
	Health promotion		21.212	2.994	0.269	7.084 **
	NCDs presence		−125.729	33.363	−0.132	−3.769 **
	Occupational hazard prevention		9.825	3.211	0.118	3.060 *
	Gender (male)	0.384	87.110	40.332	0.077	2.160 *

* *p* < 0.01, ** *p* < 0.001.

## Data Availability

The original contributions presented in this study are included in the article. Further inquiries can be directed to the corresponding authors.

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
