# Peer review of "Determinants of Quality of Life in Thai Caregivers of Dependent Older Adults: The Role of Health Promotion and Occupational Risks"

_ijerph, 2025, doi:10.3390/ijerph22040578_

Round 1

Reviewer 1 Report

Comments and Suggestions for Authors

Dear Authors,

I enjoyed reading your article, which is well written and interesting.

I have only a few minor suggestions:

- in the abstract and introduction you use the term "informal caregivers". However this cateogry includes not paid caregivers (mainly family members). Then you define them as "in home caregivers", which the correct term. My advice is to replace "informal caregivers" with "in home caregivers" throughout the text. 

Discussion:
Your data reports that most in home caregivers work for more than 8 hours a day. Are there laws in Thailand that regulate the maximum duration of the shifts?

It would be interesting to include in the discussion a comparison with healthcare workers and to specify the main differences within healthcare workers and in-home caregivers (unavailability of mechanical aids, working in pairs, regulation of shifts, and the availability of a risk assessment.

Best regards

Reviewer 2 Report

Comments and Suggestions for Authors

Introduction
This study is highly relevant and timely, given the rapid aging of Thailand’s population. The focus on informal caregivers is particularly significant, as research on this population remains limited, especially in the Asian cultural context. The study’s contribution is valuable in addressing this gap.

However, the manuscript lacks a broader international comparison. A discussion that contextualizes Thailand’s caregiving system in relation to other countries with similar socioeconomic structures would strengthen the study’s impact. By identifying both the strengths and challenges of Thailand’s caregiving system in comparison to other nations, the research could offer more comprehensive insights.

Additionally, while the research objective is clearly stated, the study does not explicitly define its research questions. If the primary aim is to analyze the determinants of HR-QOL, it would be beneficial to clearly outline the key variables of interest and the specific hypotheses being tested. Explicitly presenting the research questions would enhance clarity and improve the logical flow between the introduction and methodology sections.

The role of informal caregivers may differ across countries due to cultural variations. A more detailed description of the responsibilities of informal caregivers in Thailand (e.g., medical support, emotional care, assistance with daily activities) would provide valuable context. Currently, informal caregivers are treated as a homogenous group, but variations in roles and caregiving burdens should be considered.

Materials and Methods
The study effectively examines the impact of various independent variables (e.g., health promotion, occupational health protection, and NCD prevention) on HR-QOL. However, it remains unclear how these variables specifically influence HR-QOL. The inclusion of a conceptual framework or theoretical model (e.g., the Health Belief Model, Stress Model) would provide a stronger foundation for the study and clarify the relationships between variables. A more explicit discussion of the theoretical background would improve the coherence and rigor of the methodology section.

Results
The study presents clear findings on the factors affecting HR-QOL. However, a more in-depth discussion on the primary contributors to HR-QOL decline is needed. While the study identifies occupational risks associated with caregiving, it would be useful to compare the relative impact of these risks. Identifying which occupational hazard has the most significant effect on HR-QOL would provide more actionable insights.

Additionally, the finding that occupational hazard prevention behaviors have a relatively minor effect on HR-QOL suggests that such measures alone may not be sufficient to improve caregivers’ well-being. This raises important policy implications regarding the limitations of workplace interventions. A discussion on how occupational health strategies can be integrated with broader health promotion efforts would add value to the findings.

Discussion
One notable finding is that male caregivers reported higher HR-QOL scores than their female counterparts. However, the manuscript does not provide an explanation for this gender disparity. It would be beneficial to explore potential reasons for this difference, such as variations in caregiving workload, societal expectations, or access to social support. Including a discussion on gender-based caregiving experiences would improve the depth of the analysis.

Moreover, the study reports relatively higher scores in psychological well-being (e.g., social functioning, vitality, and mental health) despite the physical challenges faced by caregivers. This may be influenced by cultural factors such as Thailand’s collectivist society and the role of religious or familial support. A brief discussion on these protective factors would strengthen the interpretation of the results.

While the study highlights the importance of health promotion programs, it does not provide concrete recommendations on how such programs should be implemented. Citing successful caregiver support programs from other countries (e.g., South Korea’s structured caregiver training initiatives) could offer practical guidance for policymakers. Providing specific examples of effective interventions would enhance the study’s applicability to real-world settings.
